# Differential Expression, Tissue-Specific Distribution, and Posttranslational Controls of Phosphoenolpyruvate Carboxylase

**DOI:** 10.3390/plants10091887

**Published:** 2021-09-13

**Authors:** Lorrenne Caburatan, Joonho Park

**Affiliations:** 1Department of Fine Chemistry, Seoul National University of Science and Technology, Seoul 01811, Korea; lorrennecaburatan@gmail.com; 2Department of Nano Bio Engineering, Seoul National University of Science and Technology, Seoul 01811, Korea

**Keywords:** phosphoenolpyruvate carboxylase, single-cell C_4_ plant, differential expression, tissue-specific expression, posttranslational modification, phosphorylation, monoubiquitination

## Abstract

Phosphoenolpyruvate carboxylase (PEPC) is a ubiquitous cytosolic enzyme, which is crucial for plant carbon metabolism. PEPC participates in photosynthesis by catalyzing the initial fixation of atmospheric CO_2_ and is abundant in both C_4_ and crassulacean acid metabolism leaves. PEPC is differentially expressed at different stages of plant development, mostly in leaves, but also in developing seeds. PEPC is known to show tissue-specific distribution in leaves and in other plant organs, such as roots, stems, and flowers. Plant PEPC undergoes reversible phosphorylation and monoubiquitination, which are posttranslational modifications playing important roles in regulatory processes and in protein localization. Phosphorylation activates the PEPC enzyme, making it more sensitive to glucose-6-phosphate and less sensitive to malate or aspartate. PEPC phosphorylation is known to be diurnally regulated and delicately changed in response to various environmental stimuli, in addition to light. PEPCs belong to a small gene family encoding several plant-type and distantly related bacterial-type PEPCs. This paper provides a minireview of the general information on PEPCs in both C_4_ and C_3_ plants.

## 1. Introduction

Phosphoenolpyruvate (PEP) carboxylase (PEPC) is a cytosolic enzyme involved in the irreversible β-carboxylation of PEP in the presence of HCO_3_^−^ to yield oxaloacetate and inorganic phosphate (Pi) [1]. PEPCs play crucial roles in a number of metabolic functions related to plant growth and development [2], such as carbon metabolism [3] and maintenance of cellular pH balance in opening stomata. These enzymes also contribute to the accumulation of organic acids in fruits, in the supply of organic acids that are exuded from roots, and in Pi leaching to soil for aluminum detoxification [4,5].

Two classes of plant PEPCs have been identified in a number of studies [6,7]. Class-1 PEPCs are homotetramers composed of plant-type PEPC (PTPC) polypeptides of approximately 100–110 kDa, while Class-2 PEPCs are hetero-octameric molecules, which include 116–118-kDa bacterial-type PEPC (BTPC) polypeptides. PTPCs generally have a conserved N-terminal serine phosphorylation domain and can be either photosynthetic, such as in C_4_ and crassulacean acid metabolism (CAM) plants, or non-photosynthetic, such as in C_3_ plants [1,8]. In this review, we focus on characteristics and posttranslational controls of C_4_-plant PEPCs, with information on higher-plant PEPCs provided for comparison.

## 2. Differential Expression of PEPCs at Different Stages of Plant Development

In a study by Westhoff and Gowik [9], differential gene expression of PEPC enzymes was observed in both C_4_ and C_3_ species of the genus *Flaveria*. A sequence alignment with the maize enzyme [10] showed the presence of a lysine residue at position 347 of the PEPC isoenzymes in both maize and the C_4_ species *F. trinervia*, while arginine was found at this site in the C_3_ species *F. pringlei* and in the C_3_–C_4_ intermediate species *F. pubescens*. Moreover, at position 774, C_4_ *Flaveria* species have a serine residue, while C_3_ and C_3_–C_4_ intermediate plants contain alanine [9].

Accordingly, the differential expression of genes encoding C_4_-cycle enzymes, such as PEPC, is largely due to transcriptional control [9,11,12]. In the single-cell C_4_ species *Bienertia sinuspersici*, differential expression of PEPC isozymes has been observed at different developmental stages [13]. Analysis of relative mRNA expression of PEPC isoforms in extracts from young, intermediate, and mature leaves of *B. sinuspersici* showed that *PEPC3* was predominantly expressed in the early stage, while *PEPC1* and *PEPC2* were expressed in the intermediate and mature stages of leaf development, respectively (Appendix A). Another type of the PEPC isozyme, PEPC4, was also highly expressed in young leaves.

Sequence alignment of the former three PEPC isoforms showed that PEPC3 contained an arginine residue at position 346 and alanine at position 774, which is believed to be typical for a non-photosynthetic C_3_ PEPC. The suggestion that PEPC3 is a non-photosynthetic enzyme is consistent with the data of Voznesenskaya and colleagues [14], who confirmed the presence of Rubisco as a potential photosynthetic agent in the early stage of leaf development in *B. sinuspersici*. This information was further verified by Caburatan et al. [13], who showed that upon differentiation into the peripheral compartment (PC) and central compartment (CC), Rubisco in mature leaves of *B. sinuspersici* highly accumulated in the CC, which has more primitive cells than those in the PC, right before the compartmentalization occurred.

Conversely, PEPC2 harbors lysine and serine residues at amino acid positions 346 and 774, respectively, and is believed to be characteristic of C_4_ photosynthesis. Meanwhile, PEPC1 has serine and alanine residues at sequence positions 346 and 744, respectively, which are considered to be determinants of C_3_–C_4_ intermediate photosynthesis [15]. The transition from C_3_ photosynthesis in the early stage to C_4_ photosynthesis in the mature stage was verified in a study that showed that PEPC and PPDK, which are both involved in primary steps of C_4_ photosynthesis, were highly expressed in the PC, the compartment that occurs later in the development, when C_4_ photosynthesis fully develops. Thus, it can be recognized that in a single-cell C_4_ plant, such as *B. sinuspersici*, there is only one type of chloroplasts in control of photosynthesis during the early stage, when this single mass starts to differentiate into an intermediate leaf, which eventually creates two types of chloroplasts in the mature leaf. The single type of chloroplasts transitions from C_3_-like photosynthesis until differentiation into dimorphic chloroplasts, which perform C_4_ photosynthesis [16]. The utilization of C_3_ photosynthesis in the young leaf of a single-cell C_4_ plant has been confirmed by a relatively high amount of Rubisco in chloroplasts, which satisfies the need for C_3_ photosynthesis. A change in the PEPC expression occurs at the same time as the photosynthetic pathway transitions to a C_3_–C_4_ intermediate pathway, up until it reaches the stage of the C_4_-type photosynthesis pathway, which is utilized by dimorphic chloroplasts. This differential gene expression observed at different developmental stages in the single-cell C_4_ plant *B. sinuspersici* may imply a shift in a metabolic or photosynthetic pathway in chloroplasts, which is supported by PEPC molecules after achieving a certain stage of leaf development. Similar to PEPC3 and PEPC1, PEPC4 has alanine at position 774, which is typical of non-photosynthetic PEPCs (Figure 1).

Regarding C_3_ PEPCs, a study by Blonde and Plaxton [17] detected two types of PEPCs, PEPC1 and PEPC2, in castor oil seeds (COS). PEPC1 is a p107 homotetramer, while PEPC2 is an unusual 681-kDa hetero-octamer, composed of p107, as in PEPC1, but supplemented with a structurally and immunologically unrelated p64 [17]. In the case of COS, immunoblot analysis showed that the PEPC expression differed at different developmental stages, including a heart-shaped embryo, midcotyledon, full cotyledon, early maturation stage, and full maturation stage. The results demonstrated that the overall PEPC activity was relatively high until the full cotyledon stage of the developing COS endosperm, and in-gel activity staining indicated that the PEPC1/PEPC2 ratio progressively increased. However, only PEPC1 was detected in the early maturation and full maturation stages of developing COS, providing another example of differences in PEPC expression or activity at various developmental stages.

Other studies of PEPCs showed their differential sensitivities to a low pH environment and to malate [18]. Two varieties of grape berry were found to be more sensitive to a low pH and malate during the ripening stage than in earlier stages of development, suggesting participation of PEPC in the control of malic acid accumulation. A study by Or and colleagues [19] also reported a higher accumulation of PEPCs during early stages of berry development, followed by a decline at a later stage and reinduction during ripening, which suggests developmental PEPC regulation to control malate metabolism throughout berry development.

## 3. Tissue-Specific Distribution of C_4_ and C_3_ PEPCs

PEPC distribution can vary in different tissues and in specific areas of a particular tissue. As in the case of PEPC4 of *B. sinuspersici*, a 107-kDa PEPC (p107) was detected on immunoblots of extracts from young, intermediate, and mature leaves and was found to be more apparent in mature leaf samples [13].

Furthermore, immunoblotting of extracts obtained from three regions of mature leaves, namely, the base, middle part, and tip, showed that the p107 PEPC polypeptide was highly expressed in the tip portion of the mature leaf but not in the base and middle portion [13]. These findings could be due to an earlier maturation of the tip portion of the leaf than that of its lower part, indicating that PEPC expression is higher in mature leaves than in earlier ones.

A study by Penrose and Glick [20] showed that the level and activity of the PEPC protein were moderate and relatively constant in leaf shoots of sorghum, another C_4_ plant, throughout a 45-day period. Conversely, it was found that the PEPC protein attained a high level in sorghum roots, which peaked at approximately 14 days after seed germination, suggesting that the PEPC present in roots may play a significant role in plant development. In a further study in sorghum, Abergel and Glick [2] observed that the PEPC in leaf and shoot extracts had respective prominent bands of 103 and 101 kDa, while that in the root extract had a faint band of approximately 101 kDa. Their dot-blot experiment suggested that the PEPC mRNA levels in both tissues attained a steady state at approximately 3–9 days after germination. However, the level of a pulse-labeled PEPC was lower in shoots than in leaves, consistent with higher expression of PEPC in mature leaves. The late expression of PEPC in sorghum roots (~14 days after germination) could be due to posttranscriptional regulation of untranslatable PEPC mRNA, which might have been synthesized early in plant development and expressed later [2,20]. Further study of Ruiz-Ballesta and colleagues [21] in sorghum seeds PEPC presented four C_3_ PTPCs (SbPPC2, SbPPC3, SbPPC4, and SbPPC5) and a C4 photosynthetic PTPC (SbPPC1). SbPPC4 were detected at higher levels during the early seed development shown to be translated into p107 PTPC subunits. It is deemed to play a major role during the early stage of development known as the phase of cellularization. Conversely, SbPPC2 and SbPPC3 have higher levels of detection throughout the life cycle with SbPPC2 present as a monoubiquitinated p110 subunit in the embryo of both dry and post-imbibed seeds. SbPPC3 was the only one detected in the aleurone or endosperm and it was highly abundantly in the embryo. SbPPC5 was detected only as p110 polypeptides in stage I of sorghum seed development [21].

Tissue-specific distribution of PEPC was also observed in a castor oil plant by O’Leary and colleagues [7]. The expression levels of the *Ricinus communis* PEPC gene *RcPPC4* appeared more prominent in budding and expanding leaves than in mature ones, while those of the PEPC gene *RcPPC3* were higher in the integument and hypocotyl of the flower than in the entire male flower. It was also observed that *RcPPC3* and *RcPPC4* co-expressed in the tissues of developing seeds, flowers, and leaves, which was interpreted as a necessary interaction for the formation of a novel hetero-octameric Class-2 PEPC complex [2,7,22]. 

Sanchez and colleagues [6] identified four genes, encoding four different PEPCs, AtPPC1–4, in *Arabidopsis*. Except for AtPPC4, these molecules are known to be plant-type PEPCs, while AtPPC4 has been reported to encode a PEPC without any phosphorylation motif. Tissue specificity was demonstrated for these PEPCs based on differential expression of their corresponding genes in *Arabidopsis* organs. The *AtPPC1* and *AtPPC4* genes were expressed in both roots and flowers; the *AtPPC3* gene was only expressed in roots, while *AtPPC2* transcripts were detected in all *Arabidopsis* organs, which suggests that the latter is a housekeeping gene. The PEPC activity was the highest in the roots, in which all of the four genes were typically found to be expressed [6]. In this study, salt and drought stress conditions were applied to *Arabidopsis*, and the results showed differential induction of PEPC gene expression in the roots, with *AtPPC4* having shown the highest levels of induction in response to both stresses [6]. These findings suggested that stress can also cause differential expression of plant PEPCs in various tissues or organs. A study in the common iceplant [23] reported that salt stress led to differential expression of two PEPC isogenes during CAM induction, and stress-induced metabolic transition was accompanied by up to a 50-fold increase in the activity of PEPCs.

In a recent study of Yamamoto and colleagues, PEPC expressed in the external layer of seed coat (ELSC) during seed formation in soybean was investigated. Results in this study showed that there are likely six PEPC isogenes (*Gmppc1*, *Gmppc2*, *Gmpppc3*, *Gmpppc7*, *Gmpppc16*, and *Gmppc17*) which exhibited relatively high levels of expression in ELSC. Among these, Gmppc1 and *Gmppc7* were known to predominantly express and exhibit the highest gene expression levels among the analyzed samples. It was also found out that while *Gmppc1* and *Gmppc7* are highly expressed in the ELSC, on the contrary, the isogenes *Gmppc2* and *Gmppc3* are highly expressed in the developing whole seeds [24].

The tissue-specific distribution of PEPC gene expression in the C_3_ and C_4_ plants reviewed is summarized in Table 1.

## 4. Posttranslational Modifications (PTMs): Phosphorylation and Monoubiquitination of PEPCs

Phosphorylation and monoubiquitination are PTMs that play important roles in regulatory processes and in protein localization. Phosphorylation is a reversible PTM, which usually occurs in vascular plants at a conserved serine residue near the N-terminus of a protein [27,28] and is mediated by a calcium-insensitive PEPC kinase [29]. Plant PEPCs undergo reversible phosphorylation, which is mediated by a calcium-independent Ser/Thr protein kinase (PEPC-PK), encoded by a small multigene family in C_4_ plants [30,31]. Transcription and protein synthesis of C_4_ PEPC-PK are known to be light dependent and regulated via a cascade of complex signal transduction pathways [4,31,32]. In maize, PEPC-PK specifically phosphorylates PEPC at Ser-15, while in sorghum, PEPC is phosphorylated at Ser-8 [33]. The serine residue position has been reported to be conserved in C_4_ PEPC isoforms among various plants, while an alanine residue is present in other isoforms [32]. Replacement of serine with alanine has been observed to lower the *K*_m_ value for PEP [15].

In the C_4_ plant *B. sinuspersici*, both PTMs have been reported to be conserved in all PEPC isozymes [13]. Phosphorylation of the 107-kDa PEPC in the tip portion of the mature leaf was shown to be more prominent than that at the base or in the middle portion of mature leaves. This prominent occurrence of phosphorylation in the mature leaf of *B. sinuspersici* is crucial for a number of physiological processes, including direct control of enzymatic activity, which may affect PEPC dominance in the PC during the mature leaf stage, although a further study is needed. Phosphorylation also generates specific docking sites for protein–protein interactions [1], which may lead to further changes in the activities of intracellular enzymes. Thus, the homodimer protein 14-3-3 may cause conformational changes or influence interactions between target molecules upon binding to specific phosphorylation sites on diverse target proteins, when phosphorylation itself cannot fully drive changes in intracellular enzyme activities [34].

Moreover, PEPC enzymes are allosterically regulated by positive or negative effectors. These effectors vary in different species and exert different effects on kinetic properties of PEPCs, including among isoforms in the same organisms [8,32,34,35]. Phosphorylation activates a PEPC enzyme by making it more sensitive to glucose-6-phosphate (Glu-6-P) and less sensitive to malate or aspartate [35]. In maize, phosphorylation of a C_4_ PEPC was observed to be regulated by a thioredoxin-mediated reductive activation of PEPC-PK [33]. It is thought that redox signaling, which is likely to occur in the illuminated mesophyll cytosol, may play a significant role in the activation of PEPC-PK, thus exerting regulatory effects on the C_4_ PEPC. Other studies also reported that different photosynthetic electron flow inhibitors may block the upregulation of PEPC-PK and phosphorylation of PEPC [33]. These effects may be partly due to the lack of PEPC-PK-reducing equivalents, which, under normal conditions, are supplied by the electron transport system of thylakoid membranes in illuminated chloroplasts [33]. Other known C_4_ PEPC inhibitors are anionic phospholipids, especially phosphatidic acid (PA), which binds to C_4_ PEPC. In a study by Monreal and colleagues [31], the addition of 50 μM PA decreased the PEPC activity to 40% of the control activity in both sorghum and maize plants. Other anionic phospholipids, such as phosphatidylinositol, phosphatidylinositol-4-phosphate, and lyso-PA, also showed similar levels of inhibition of PEPC activity, while phosphatidylserine showed partial inhibition and the control lipids phosphatidylcholine and phosphatidylethanolamine showed no effects. In eukaryotic cells, PA is a minor lipid and is mostly involved in the biosynthesis of structural phospholipids and glycolipids in the endoplasmic reticulum and plastids [31]. PA acts as a secondary messenger, which can be generated by phospholipase D (PLD) in the plasma membrane when it is activated by a signal such as a pathogen elicitor, hormone binding to specific receptors, and other multiple stress-related signals [31,36,37].

This C_4_ PEPC inhibition by various phospholipid entities is direct and is deemed to be independent of the phosphorylation status or known allosteric regulators, as confirmed by the separate addition of Glu-6-P and L-malate to media with PA. The data showed that the addition of allosteric regulators and phosphorylation of the PEPC enzyme had no effect on the PEPC-PA interaction [31]. Law and Plaxton [38] found that pH and certain metabolites had effects on a banana fruit PEPC. The PEPC activity at pH 8.0 was twice of that at a neutral pH of 7.0. However, at pH 7.0, certain metabolites are known to either activate or inhibit the activity of PEPC. Among the metabolites that display pH-dependent modulation are Glu-1-P, Glu-6-P, fructose-1-phosphate, fructose-6-phosphate, and glycine-3-phosphate. These are considered positive effectors, which double the PEPC activity upon changing pH from 8.0 to 7.0. The known negative effectors are malate, succinate, aspartate, and glutamate [38]. In addition to phosphorylation, the membrane-surface charge has been suggested to possibly regulate the PEPC activity, based on in vivo studies. However, the mechanism by which the PEPC activity and cellular localization can be affected by dynamic changes in the presence of negatively charged membrane lipids needs further verification, including in vivo analysis of GSP fusions of a C_4_ PEPC protein [39]. A summarized list of known phosphorylation activators and inhibitors is presented below (Table 2). 

PEPC phosphorylation has been known to be diurnally regulated and delicately changed in response to various environmental stimuli, in addition to light [31,37]. The PEPC induction by light in C_4_ plants has been shown to be regulated at the transcriptional level [38,40]. Diurnal regulation of PEPC phosphorylation has been observed in the C_4_ *B. sinuspersici* plant. Immunoblotting of samples exposed to light showed a prominent phosphorylated band of approximately 107 kDa. C_4_ PEPCs are thus modulated by light via phosphorylation [17]. The mechanism of light modulation was somewhat clarified by Monreal and colleagues [31], who elaborated on the participation of PLD and PA in the signaling cascade of light-dependent upregulation of a sorghum leaf PEPC. It has been proposed that a light signal is the primary event that leads to the activation of phospholipase C and the synthesis of PEPC-PK in the mesophyll of C_4_ plants [41]. It was also shown that light increased the PLD activity and production of PA in sorghum leaf discs, thus supporting a link between PLD-dependent PA production and a light signal [31,42].

Avasthi and colleagues [43] elucidated, at molecular levels, the interplay between light and temperature in the in planta modulation of a C_4_ PEPC from leaves of *Amaranthus hypochondriacus*. The results of this study showed that the peak of PEPC activity, which may indicate an increase in the activation by Glu-6-P or a decrease in malate sensitivity, was always preceded by a maximum light intensity and coincided with a high temperature. This consistent trend was observed throughout the year, thus indicating an apparently strong relationship between the modulation of PEPC activity by malate/Glu-6-P and light or temperature. The PEPC activity was found to be higher during a light period when it was assayed at 0.05 mM PEP and 0.05 mM NaHCO_3_ than at considerably higher concentrations. It is possible that light illumination increases the affinity of PEPC for PEP or HCO_3_^−^, making the enzyme function more efficiently, including at a low substrate concentration [33,43,44]. 

Further studies revealed that light had a greater influence on PEPC activity, while temperature exerted a much greater effect on the regulatory properties of the C_4_ PEPC [45]. In this case, the levels of protein phosphorylation increased as the day progressed, reaching a maximum at approximately 15 h, and eventually decreased to a minimum at 24 h. These levels were higher in May than in December; thus, PEPC was phosphorylated at night in May but was almost completely dephosphorylated at night during December [43]. The maximum PEPC activity, minimum malate sensitivity, and highest activation levels by Glu-6-P at 15 h may be due to the maximum phosphorylation levels, which in turn can be attributed to higher photoactivation of PEPC at warm temperatures. The results further showed that during May, light had a greater influence on the PEPC protein levels, while temperature had a much greater effect on both the PEPC protein and phosphorylation levels. During December, however, light exerted much influence on the PEPC mRNA levels, while temperature had a much greater influence on the phosphorylation status. In this study, the variation in light or temperature in relation to protein levels, phosphorylation status, and mRNA levels in leaves of *A. hypochondriacus* was also determined on a typical day in May and December. It was concluded that the phosphorylation status and not the mRNA or protein level was more crucial for the daily and seasonal patterns of PEPC activity and regulatory properties [43].

Moreover, phosphoproteomics conducted on *Arabidospsis* rosettes under controlled CO_2_ and O_2_ mole fractions showed a slight decrease in phosphorylation of the PEPC1 isoform at high concentrations of CO_2_ and low fraction of oxygen (1000 umol mol^−1^ and 0.03% respectively). This may imply a downregulation of PEPC activity at higher levels of photosynthetic rates [45,46].

Monoubiquitination is another reversible PTM that mediates a number of protein–protein interactions by recruiting the ubiquitin-binding domain of client proteins and that mediates protein localization to help control processes such as gene expression, endocytosis, transcription and translation, DNA repair, and signal transduction [11,47,48]. The globular protein ubiquitin (UB) is highly conserved in eukaryotic cells. UB modifies target proteins by covalently attaching itself via an isopeptide bond between the C-terminal glycine of UB and the ɛ-amino acid group of a lysine residue on a target protein [48]. UB attachment to cellular proteins involves a multienzyme system consisting of activating, conjugating, and ligating enzymes [1]. Unlike PEPC phosphorylation, no diurnal effects were observed on the intensity of a monoubiquitinated band of approximately 110 kDa under the same light and dark conditions in *B. sinuspersici* leaf samples [13]. Immunoblotting of the *B. sinuspersici* PEPC from the tip part of mature leaves showed the hydrolysis of the 110-kDa band into a 107-kDa band upon treatment with a deubiquitinating enzyme. Other studies suggested that the monoubiquitinated form of PEPC is significantly more sensitive to malate and aspartate inhibition than the deubiquitinated one is [47,48]. In COS, a regulatory monoubiquitination occurs at Lys-628 of a ~110-kDa polypeptide, while in vitro, deubiquitination converts the p110:p107 PEPC heterotetramer from germinated COS into a p107 homotetramer [48]. Regulatory phosphorylation of PTPC subunits occurred at Ser-11 while the BTPC subunits occurred at Ser-425 [11]. Below is a summary of known phosphorylation and monoubiquitination sites in several known plants (Table 3).

Conversely, in germinating sorghum seeds, p107 is monoubiquitinated to form p110 at Lys-624, which is known as a conserved residue in vascular plant PEPCs and is proximal to a PEP-binding or PEPC catalytic domain [49]. Further study in sorghum seeds detected SbPPC2, SbPPC3, and SbPPC4 PTPCs in both p110 and p107 while SbPPC5 was detected only in p107. The p107 N-terminal of SbPPC3 is a non-phosphorylated Ser7, but the monoubiquitinated SbPPC3 p110 subunits were determined to be phosphorylated at Ser7. It follows that the monoubiquitinated p110 subunits of the other isoenzymes such as SbPPC2 and SbCC4 as well as the deubiquitinated p107 subunits of SbPPC5 were also phosphorylated at their conserved N-terminal seryl phosphorylation sites [21]. Hence, in the case of sorghum seeds, monoubiquitination and phosphorylation occur simultaneously on the same PTPC polypeptide and are therefore not mutually exclusive as in the case of other plant PTPCs [21]. 

Moreover, in a study of PEPC in ammonium-stressed sorghum, the expression of the C_3_-type PPC5 and the bacterial-type PPC6 was not detected in leaves and in roots but NH4^+^ has been shown to increase the phosphorylation state of PEPC and PEPC-kinase activity in the leaves both in the presence and absence of light [50]. NH4^+^ also increased the amount of monoubiquitinated PEPC in sorghum roots; hence, in such a case, although the roles of phosphorylation and monoubiquitination seem to be opposite in manner, both PTM’s co-exist in leaves and in roots, respectively. An increase in phosphorylation was a consequence of favorable *PPCK1* expression in leaves, while PEPC activity in roots augmented due to the increase in *PPC3* expression thus increasing the capacity of sorghum to cope with the ammonium stress [50]. Recent studies of Gandullo and colleagues [51] on sorghum leaves found that phosphorylated PEPC is less sensitive to proteolysis than its dephosphorylated form in vivo and in vitro. C_4_-PEPC proteolysis in the presence of pC19 (which is a synthetic peptide containing the last 19 amino acids from the C-terminal end of the PEPC subunit) was prevented by the PEPC allosteric effector glucose 6-phosphate (Glc-6P) and by PEPC phosphorylation [51]. Thus, in such a case, PEPC phosphorylation renders protection from degradation by cathepsin proteases present in semi-purified PEPC fraction (with protease activity) than the dephosphorylated form [51]. Another study in sorghum leaves also claimed that salt-induced nitric acid synthesis protects photosynthetic PEPC activity from oxidative inactivation while promoting phosphorylation. In such a case, phosphorylation is thought to further guarantee optimal function in suboptimal conditions [52]. 

In the case of C_3_ plants, a study by Fukayama et al. [53] found nocturnal phosphorylation in three hygrophytic monocots, including rice, *Monochoria vaginalis*, and *Sagittaria trifolia*. In these plants, the phosphorylation levels at night were higher than those during the day. Among 24 other C_3_ species studied, 12 species showed diurnal phosphorylation, and 12 showed almost no changes in the phosphorylation levels between the day- and nighttime [53]. Two of the three monocots, rice and *M. vaginalis*, have a PEPC protein inside the chloroplast in addition to the cytosol, which suggests a relationship between the presence of the chloroplastic PEPC and nocturnal phosphorylation of the cytosolic enzyme [53]. 

## 5. PEPC Complexes: PTPCs and BTPCs

Plant PEPCs belong to a small gene family encoding several PTPCs and distantly related BTPCs [7,22]. The Class-1 PEPC of developing COS is a known 410-kDa homotetramer, composed of identical p107 subunits, which are phosphorylated in vivo [48]. The mechanisms regulating Class-1 PEPC activity in different plant tissues under various physiological conditions include phosphorylation, monoubiquitination, changes in intracellular pH, and allosteric effectors. In the castor oil plant, monoubiquitination is more widespread than phosphorylation and appears to be a more predominant PTM of the Class-1 PEPC [7,34]. Conversely, the Class-2 PEPC of developing COS is a novel 910-kDa hetero-octameric complex in which the same PTPC subunits as in the Class-1 PEPC isozyme are tightly associated with four BTPC subunits. It has been hypothesized that Class-2 PEPCs function as a metabolic overflow mechanism that is able to maintain a significant flux from PEP to L-malate under physiological conditions, which will inhibit Class-1 PEPCs [7,54]. The COS Class-2 PEPC has also been hypothesized to support a significant flux of PEP to malate, which is needed for fatty acid synthesis, dominating the developing COS leucoplast metabolism. Within the Class-2 PEPC complex, both PTPC and BTPC subunits exhibit PEPC activity in developing COS [7]. The difference is that the PTPC subunits have a higher affinity for PEP and are more allosterically sensitive than BTPCs, which are remarkably insensitive to PTPC inhibitors, such as L-malate and L-aspartate [54].

In *B. sinuspersici*, another type of PEPC, termed PEPC4, was found to be highly expressed in young leaves (unpublished data), in addition to PEPC3 [13]. PEPC4 is believed to be the BTPC counterpart of Class-2 PEPCs because of its presence in young leaf tissue. According to O’Leary et al. [7], upregulation of BTPC and thus Class-2 PEPC expression appear to be distinctive features of rapidly growing and/or biosynthetically active tissues; the latter requires a large anaplerotic flux of PEP to replenish malate and other tricarboxylic acid cycle intermediates, which are consumed via anabolism. Thus, BTPC subunits may facilitate the accumulation of a pool of organic acids, which can either generate osmotic potential and/or serve as sources of C-skeletons and reducing power to support anabolism [7]. In the case of *B. sinuspersici*, the BTPC component is thought to be a 117-kDa tetraenzyme (Appendix A; unpublished data). A graphical representation of the possible association of Class-1 and Class-2 PEPCs with *B. sinuspersici* leaf development stages is shown in Figure 2.

While the hetero-octameric Class-2 PEPC is highly expressed in young leaves of *B. sinuspersici*, the Class-1 PEPC is expressed in both intermediate and mature leaves, based on the presence of the phosphorylated p107 and monoubiquitinated p110 PTPCs. The close interaction between BTPCs and PTPCs in a Class-2 PEPC complex during early leaf development may indicate that BTPCs have a significant regulatory role that is commenced during the early stage but is no longer needed later, as other mechanisms, such as PTMs, can provide further regulation at a more advanced stage of leaf development.

Hence, from a single cell during the early leaf development of *B. sinuspersici*, BTPC acts as a regulatory enzyme, which provides components that may be critical for PEPC catalysis and heterologous expression, such as in green algae, or for PEPC activity, such as in the case of a COS BTPC [54,55]. However, as leaf development progresses to intermediate and mature stages, including the development of dimorphic chloroplasts, PTPCs of the Class-1 PEPCs become dominant and are allosterically regulated or undergo PTMs, such as phosphorylation and monoubiquitination.

This activation of Class-1 PEPCs by phosphorylation has mainly been observed in plant tissues in which a high and tightly controlled flux of PEP to malate plays a well-known metabolic role, e.g., upon nutritional Pi starvation during CO_2_ assimilation in C_4_ and CAM leaves, during rapid nitrogen assimilation following NH_4_^+^ or NO_3_^−^ resupply to nitrogen-limited cells, or during photosynthate partitioning to storage end-products in developing COS [1,56,57,58,59].

Moreover, in the case of sorghum seeds, *SbPPC6* genes encodes the distantly related BTPC. However, transcript of this gene was low during the development and germination of seeds [21].

In C_3_ plants, such as lily in a study by Igawa and colleagues [60], a BTPC, ubiquitinated PTPC, and PTPC (LlBTPC:Ub-LlPTPC:LlPTPC) complex was observed to be formed in the vegetative cell cytoplasm during late pollen development. In this study, BTPCs were significantly expressed in the pollen of both lily and *Arabidopsis*, but there was a difference in the initiation of the expression of each BTPC during pollen development. LIBTPC started to accumulate following GC formation after pollen mitosis I, whereas AtBTPC expression started immediately following sperm cell formation after pollen mitosis II [60]. Meanwhile, the lily PTPC and monoubiquitinated LlPTPC remained at constant levels during pollen development. In the late bicellular pollen, LlBTPC formed a hetero-octameric Class-2 PEPC complex with LlPTPC to increase the PEPC activity [60]. 

## 6. Concluding Remarks

PEPC plays a crucial part in plant carbon metabolism. Different stages of plant development portray differential expressions of PEPC activity as in the case of *B. sinuspersici* leaves, castor oil plant leaves, sorghum seeds, *Arabidopsis* root and rosettes, and soybean seeds. This differential expression can also be tissue specific, and at large, varies depending on plant organs or tissues. Various PEPC isoforms may also differ in PEPC activity depending on conditions either in vivo or in vitro. Factors such as salinity, light, temperature, and ammonium-stress may also regulate PEPC activity by simply affecting post-translational control mechanisms such as phosphorylation and/or monoubiquitination. Although phosphorylation and monoubiquitination seem to be exclusive and work in an opposite pattern, studies in other plants such as in the case of sorghum roots showed that both PTMs occur simultaneously to increase PEPC activity. Both PTMs regulate Class-1 PEPC activity in different plant tissues under various physiological conditions. Other regulating mechanisms also include changes in intracellular pH, and presence of allosteric effectors. However, in the castor oil plant, monoubiquitination is more widespread than phosphorylation and appears to be a more predominant PTM of the Class-1 PEPC. While most PEPCs encode PTPCs, inclusion of the distantly related BTPCs into the Class-1 PEPCs will comprise Class-2 PEPC which was regarded as a 910 kDa hetero-octameric complex in castor plant. Within the Class-2 PEPC complex, both PTPC and BTPC subunits exhibit PEPC activity in developing COS. With the vast potential of regulatory mechanism affecting PEPC activity, an understanding of the intricate interaction between these post-translational mechanisms as well as the occurrence and function of such in different plant tissues in various stages of development both in normal and stressed conditions will be a challenge worthy to notice for one to fully describe the roles and metabolic control of plant PEPC. Such an undertaking, if considered, may further elucidate the crucial role of PEPC in plant carbon metabolism.

## Figures and Tables

**Figure 1 plants-10-01887-f001:**
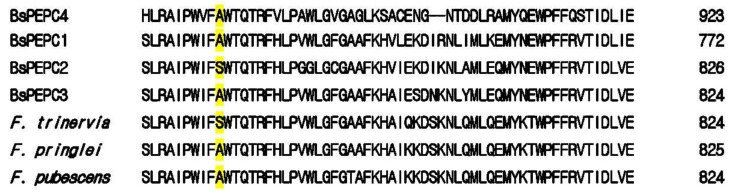
Sequence alignment of *Bienertia sinuspersici* phosphoenolpyruvate carboxylase (PEPC) isozymes (PEPC1–4) with those from *Flaveria* species and their characteristic amino acids at position 774 (yellow highlighted). Note that the sequences at this position are identical for the C_4_ species *F. trinervia* and BsPEPC2, with the amino acid serine, while the C_3_ species *F. pringlei* and C_3_–C_4_ intermediate species *F. pubescens*, which have alanine, are aligned with BsPEPC1 and BsPEPC3, which are thought to be C_3_ and C_3_–C_4_ intermediate PEPCs, respectively. The bacterial-type PEPC counterpart of *B. sinuspersici* (BsPEPC4) contains serine. The amino acid numbering follows that of PEPC3.

**Figure 2 plants-10-01887-f002:**
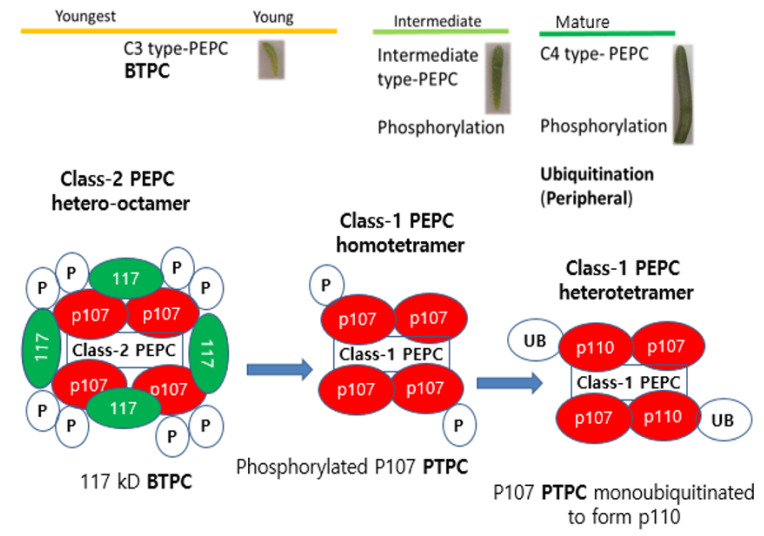
Potential working scheme for *Bienertia sinuspersici* phosphoenolpyruvate carboxylases (PEPCs). Class-2 hetero-octameric PEPCs, associated with bacterial-type PEPCs (BTPCs), are potentially expressed during the early stage of leaf development, while Class-1 homotetrameric PEPCs are expressed during the intermediate and mature stages of leaf development and include phosphorylated (P) p107 PTPC and monoubiquitinated (UB) p110 PTPC subunits.

**Table 1 plants-10-01887-t001:** Distribution of PEPC gene expression in various plant organs.

Plant	Tissue/Organ	PEPC Gene(s)	Reference(s)
*Bienertia sinuspersici*	Young leaf	*PEPC3, PEPC4*	[13]
	Intermediate leaf	*PEPC1*	[13]
	Mature leaf	*PEPC2*	[13]
*Sorghum bicolor*	Mature leaf	*CP46, CP21*	[25,26]
	Shoot Root Seed	*CP21, CP28*	[26] [25,26] [15]
		*CP21, CP28* *SbCC1, SbCC2, SbCC3, SbCC4* *SbCC5, SbCC6*	
*Ricinus communis*	Budding leaf	*RcPPC4*	[7]
	Expanding leaf	*RcPPC4*	[7]
	Flower integument	*RcPPC3*	[7]
	Flower hypocotyl	*RcPPC3*	[7]
*Arabidopsis thaliana*	Root	*AtPPC1, AtPPC2, AtPPC3, AtPPC4*	[17]
	Flower	*AtPPC1, AtPPC2, AtPPC4*	[17]
*Glycine max*	Seed	*Gmppc1, Gmppc7* *Gmppc2, Gmppc3*	[24]

**Table 2 plants-10-01887-t002:** Lists of phosphorylation activators and inhibitors.

Phosphorylation Activators	Phosphorylation Inhibitors
Glucose-6-phosphate	Malate
Glucose-1-phosphate at pH 7	Aspartate
Fructose-6-phosphate at pH 7	Succinate
Fructose-1-phosphate at pH 7	Glutamate
Glycine-3-phosphate at pH 7	Anionic phospholipids (phosphatidylinositol,
Salt induced nitric acid	Phosphatidylinositol-4-phosphate, lyso-PA)

**Table 3 plants-10-01887-t003:** List of phosphorylation and monoubiquitination sites of several known plants.

	Phosphorylation Site	Monoubiquitination Site
Castor oil seeds	Ser-11 (PTPC), Ser-425 (BTPC)	Lys-628
Sorghum seeds	Ser-7	Lys-624
Maize	Ser-15	----

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
