# Peer review of "Differential Expression, Tissue-Specific Distribution, and Posttranslational Controls of Phosphoenolpyruvate Carboxylase"

_plants, 2021, doi:10.3390/plants10091887_

Round 1
Reviewer 1 Report
In this article, Caburatan and coworkers presented a review on phosphoenolpyruvate carboxylase (PEPC), a key enzyme crucial for plant carbon metabolism. Specifically, they summarized the differential expression, tissue-specific distribution, and posttranslational controls of PEPC. Overall it is a well-written paper. Although the authors addressed the phosphorylation of PEPC, other PTMs were not mentioned much. For example the roles of ubiquitination have been well documented recently (e.g., https://pubmed.ncbi.nlm.nih.gov/21841182/; https://academic.oup.com/jxb/article/67/11/3523/2197831; https://pubmed.ncbi.nlm.nih.gov/28431276/), but they were did not discussed in this manuscript. It would be great if the authors can mention those as well.Author Response
Additional informations were added with regards to some other details on monoubiquitination.
Reviewer 2 Report
I expected in a Review/minireview a final section including concluding remarks. Additionally I noted an absence of Current references in the consulted bibliography.
Although the theme is important in Plant Sciences, these changes are necessary before acceptance.
Author Response
Current references related to the review were added as well as a concluding remarks section.
Reviewer 3 Report
This study is aimed to review the differential expression, tissue-specific distribution, and posttranslational controls of
phosphoenolpyruvate carboxylase. The review is designed in an acceptable level. The work contains some helpful information that can be publsihed after revisions.
Some suggestions:
L167: Give a summarizing table or figure for this section.
L279: in vitro - in italic
I miss a conclusion section and a future prospect section at the end of the review.
Author Response
Corrections on italicization has been done. A concluding remarks section had been written as well as a prospect for further studies. Two tables (summarizing the phosphorylation and monoubiquitination sites in some plants and presenting the activators and inhibitors of phosphorylation) had been added in the fourth section.
Round 2
Reviewer 2 Report
Authors addressed properly the comments. The paper must be accepted in its present form.